# Bladder Cancer Immunotherapy by BCG Is Associated with a Significantly Reduced Risk of Alzheimer’s Disease and Parkinson’s Disease

**DOI:** 10.3390/vaccines9050491

**Published:** 2021-05-11

**Authors:** Danielle Klinger, Brian L. Hill, Noam Barda, Eran Halperin, Ofer N. Gofrit, Charles L. Greenblatt, Nadav Rappoport, Michal Linial, Hervé Bercovier

**Affiliations:** 1Department of Biological Chemistry, Institute of Life Sciences, The Hebrew University of Jerusalem, Jerusalem 91904, Israel; danielle.klinger@mail.huji.ac.il; 2Department of Computer Science, University of California Los Angeles, Los Angeles, CA 90095-1596, USA; blhill@g.ucla.edu (B.L.H.); eranhalperin@gmail.com (E.H.); 3Clalit Research Institute, Innovation Division, Clalit Health Services, Ramat-Gan 6578898, Israel; noambard@gmail.com; 4Department of Biomedical Informatics, Harvard Medical School, Boston, MA 02115, USA; 5Department of Urology, Hadassah University Hospital, Jerusalem 91904, Israel; ogofrit@gmail.com; 6Department of Microbiology and Molecular Genetics, Faculty of Medicine, The Hebrew University of Jerusalem, Jerusalem 91904, Israel; charlesg@ekmd.huji.ac.il (C.L.G.); hb@cc.huji.ac.il (H.B.); 7Department of Software and Information Systems Engineering, Faculty of Engineering Sciences, Ben-Gurion University of the Negev, Beer Sheva 84105, Israel; nadavrap@bgu.ac.il

**Keywords:** Bacillus Calmette–Guérin, tuberculosis, immunotherapy, non-muscle-invasive bladder cancer (NMIBC), neurodegenerative disease

## Abstract

Bacillus Calmette–Guerin (BCG) is a live attenuated form of *Mycobacterium bovis* that was developed 100 years ago as a vaccine against tuberculosis (TB) and has been used ever since to vaccinate children globally. It has also been used as the first-line treatment in patients with nonmuscle invasive bladder cancer (NMIBC), through repeated intravesical applications. Numerous studies have shown that BCG induces off-target immune effects in various pathologies. Accumulating data argue for the critical role of the immune system in the course of neurodegenerative diseases such as Alzheimer’s disease (AD) and Parkinson’s disease (PD). In this study, we tested whether repeated exposure to BCG during the treatment of NMIBC is associated with the risk of developing AD and PD. We presented a multi-center retrospective cohort study with patient data collected between 2000 and 2019 that included 12,185 bladder cancer (BC) patients, of which 2301 BCG-treated patients met all inclusion criteria, with a follow-up of 3.5 to 7 years. We considered the diagnosis date of AD and nonvascular dementia cases for BC patients. The BC patients were partitioned into those who underwent a transurethral resection of the bladder tumor followed by BCG therapy, and a disjoint group that had not received such treatment. By applying Cox proportional hazards (PH) regression and competing for risk analyses, we found that BCG treatment was associated with a significantly reduced risk of developing AD, especially in the population aged 75 years or older. The older population (≥75 years, 1578 BCG treated, and 5147 controls) showed a hazard ratio (HR) of 0.726 (95% CI: 0.529–0.996; *p*-value = 0.0473). While in a hospital-based cohort, BCG treatment resulted in an HR of 0.416 (95% CI: 0.203–0.853; *p*-value = 0.017), indicating a 58% lower risk of developing AD. The risk of developing PD showed the same trend with a 28% reduction in BCG-treated patients, while no BCG beneficial effect was observed for other age-related events such as Type 2 diabetes (T2D) and stroke. We attributed BCG’s beneficial effect on neurodegenerative diseases to a possible activation of long-term nonspecific immune effects. We proposed a prospective study in elderly people for testing intradermic BCG inoculation as a potential protective agent against AD and PD.

## 1. Introduction

Bacillus Calmette–Guérin (BCG), a live attenuated form of *Mycobacterium bovis*, was developed 100 years ago as a vaccine against tuberculosis (TB) [1]. BCG is administered intradermally at birth and has been shown to protect against TB, with an effect that lasts between 15 and 20 years [2,3]. Following BCG vaccination, changes in the epigenetic signature, transcription, and hematopoietic cell function [4] are evident. These alterations reflect the BCG-induced trained innate immunity [5,6]. Specifically, changes in bone marrow-derived hematopoietic stem and progenitor cells (BM-HSPCs) and monocytes lead to an increase in cytokine secretion (e.g., IL-1β, TNFα, MCP-1, and IL-8) [7], and the activation of lymphocytes, monocytes, neutrophils [8], and natural killer (NK) cells [6]. Besides, BCG induces the expansion of antigen-presenting cells of the adaptive immune response (e.g., dendritic and T-cells) [5,9].

Beyond the long-lasting use of BCG as a vaccine against TB, it is an approved immunotherapy for high-risk, non-muscle invasive bladder cancer (NMIBC) patients, used for preventing relapse in carcinoma in situ (CIS) of the bladder. About 75% of the bladder cancer (BC) patients are diagnosed with NMIBC and one third are treated with a BCG instillation protocol of which about two-thirds of the patients stay disease-free [10,11]. Following completion of the BCG treatment with the repeated instillation protocol, about two-thirds of the patients remain disease-free. While the exact mechanism underlying the protective effect of BCG is still unknown, several studies have suggested that it stimulates the Th1 response that leads to cytokine secretion (e.g., IL-2, IL-8, IL-18, and TNFα), which activates a cytotoxic response toward the remaining cancer cells [10,12]. However, understanding the biology of BCG and its effect on the bladder remains fragmented and incomplete. Over the past decades, the BCG treatment of NMIBC patients is used worldwide. Following a transurethral resection of bladder tumor (TURBT) procedure, about a third of all patients are treated with BCG [13].

Results from Alzheimer’s disease (AD) examining humans and animal models argue that immune system dysregulation contributes to the disease’s progression. AD is the primary cause of dementia in the elderly. It is a progressive neurodegenerative disorder that primarily causes loss of memory and ultimately leads to death [14]. AD affects 50 million people worldwide, and no curative treatment has been found [15,16,17]. AD pathogenesis includes the accumulation of insoluble forms of Aβ plaques, tau protein hyperphosphorylation forming neurofibrillary tangles, oxidative stress, and sustained inflammation [18]. The Aβ aggregates induce the activation of microglia and astrocytes that consequently cause sustained brain inflammation [19,20,21]. AD cases linked to ApoE4 are mostly in patients younger than 65 years old and are of a different etiology [22] than late-onset AD. The population that is investigated in this work does not encompass this aspect of AD, as the vast majority of BC patients are >70 years old [13]. In any case, it becomes evident that AD has multiple clinical characteristics that are poorly defined. Specifically, the variation in the tau pathology is composed of at least four sub-types, implicating new categories in the stages of AD [23]. In addition, differences in the clinical manifestation of AD are impacted by behavioral and lifestyle factors (e.g., loneliness, diet, and diabetes), advanced age, and additional familial risk factors besides the ApoE4 genetics [24]. All of these factors increase the difficulty to properly evaluate the efficacy of any intervention. Drugs targeting Aβ plaques and neurofibrillary tangles have not yet proved their efficacy in treating AD [25]. We anticipate that a study on a population level with a heterogeneous genetic background and diverse behavioral and lifestyle could potentially neutralize the effects attributed to uncontrolled variables. Recent evidence suggests that changes in peripheral and central adaptive immunity occur in AD patients. Nevertheless, anti-inflammatory drugs have produced disappointing results in AD treatment [26].

A growing body of publications postulates that the impact of intravesical BCG application NMIBC patients is mediated through the effect of BCG on peripheral and central immunity [13,27,28,29]. Based on preliminary results that tested AD diagnosis from bladder cancer (BC) patients treated by BCG [30], and the assumed immune involvement in the underlying mechanism of neurodegenerative disorders, we hypothesize that nonspecific immunity by BCG therapy might hold a protective capacity toward some neurodevelopmental diseases’ development.

AD incidence increases with age. Still, the median age of BC diagnosis precedes that of AD incidence, and NMIBC patients have good prospects of long-term survival. Therefore, the BC population treated or not treated by BCG is an appropriate cohort for testing our hypothesis toward the major age-dependent neurodegenerative diseases that have an underlying immune component—AD [15,31,32] and Parkinson’s disease (PD). In this study, we proposed the evaluation of the association between BCG treatment and the risk of developing AD and PD in three retrospective cohorts from Israel and California by comparing BC patients treated with BCG instillations following TURBT, with BC patients not treated with BCG. We hypothesized that BCG favorably modulates the immune system that may promote neurogenesis, and that it exhibits neuroprotective effects, eventually leading to an overall decreased risk of AD and PD in the treated cohort. To test our hypothesis, we determined if repeated exposure to BCG during the treatment of NMIBC is associated with a reduced risk of developing AD and PD. We further tested whether our hypothesis held for nonneuronal, nondegenerative events of stroke and the chronic metabolic disease of Type 2 diabetes (T2D).

## 2. Methods

### 2.1. Source of Data and Ethics

CHS: The electronic health records from Clalit Health Services (CHS). Clalit is Israel’s largest provider healthcare organization (covering ~ 4.7 million members). Data collected from 2000 to 2020 were scrutinized. Health care provider switching rates are low (1% annually), enabling the longitudinal follow-up of patients [33]. CHS databases contain both biomedical and claims data.

HUH: Electronic health records since 2000 from Hadassah University Hospitals (HUH) were examined. HUH serves approximately 1.5 million people. The Hospital has a low attrition rate, allowing for long-term studies on the treated population.

UCLAH: The UCLA Health System (UCLAH) is an academic medical provider that includes two hospitals and 210 primary and specialty outpatient locations throughout the Los Angeles area. De-identified electronic health records were extracted from the Discovery Data Repository, which contains longitudinal clinical electronic records for more than 1.5 million patients since March 2013. 

Helsinki approvals for this retrospective research on de-identified patients were obtained from the IRB of Clalit Health Services (CHS, #0160-19-COM), Hadassah University Hospitals (HUH, #0037-17-HMO), and the UCLA Health System (UCLAH, DDR agreement of de-identified data repository).

### 2.2. Cohort Definitions

In this study, we analyzed three large cohorts that differed in location, medical system, and health policies. Data from CHS include all BC patients over 60 years old and diagnosed since 2000. The ICD-9-CM code of 188.9 was used to define BC patients. Most patients included were diagnosed at BC stages T0 and T1 to meet the criteria of NMIBC (>90%, Table 1). Patients were classified as having stage T ≥ 2 BC if they received BC chemotherapy treatment (ICD-9-CM code 57.7) or underwent radical cystectomy (ICD-9-CM code 57.9). Information regarding new-onset AD in patients was defined using ICD-9-CM code 331.0 and new-onset Dementia by using ICD-9-CM code 294.20. The CHS datasets provide personalized information on age, sex, death date, BC diagnosis date, BCG administration date, dates of each BCG treatment, AD diagnosis date, stroke diagnosis date (ICD-9-CM code 434.91), PD diagnosis date (ICD-9-CM code 332.0), and mitomycin C (MMC) administration date.

The second cohort, HUH, includes 1371 BC patients [30]. Of these, 878 were treated with BCG. The HUH dataset contains data on age, gender, year of death, year of BC diagnosis, BCG administration year, and year of AD diagnosis for each patient.

The third cohort, the UCLAH dataset, includes BC patients over 60 years old and diagnosed with BC since 2013. The ICD-10 code of C67 was used, in addition to the ICD-9-CM code of 188.9 for the purpose of BC diagnosis. New-onset AD in patients was defined using ICD-9-CM code 331.0 or ICD-10 code G30, and new-onset dementia was defined using either ICD-9-CM code 294.20 or ICD-10 codes F00, F03, or F02.8. The initial UCLAH dataset includes 4794 BC patients, and among these patients, 451 were treated with BCG.

The UCLAH datasets report on age, sex, death date, BC diagnosis date, dates of all BCG treatments, AD diagnosis date, stroke diagnosis date (ICD-10 code I63), PD diagnosis date (ICD-10 code G20), and MMC administration date for each patient.

### 2.3. Inclusion–Exclusion Criteria

The different cohorts used in this study were refined by applying specialized inclusion–exclusion criteria according to the unique features in each of the dataset resources. Patients treated with both MMC and BCG or diagnosed with AD, prior to the BC diagnosis, were excluded from the cohort. The number of BCG instillations was calculated for each patient. For study purposes, patients were qualified as “BCG cases” if they were treated with ≥3 instillations of BCG within a 120-day period. This is in accordance with the gold standard of BCG treatment and instillations protocols [34] for triggering a sufficient immune response. BC patients not included in the “BCG cases” group were considered as controls.

The index date was set as the time of the third BCG instillation for the BCG cases. For non-BCG receivers, the index date was set to BC diagnosis + 92 days, to match the median time between BC diagnosis and the third BCG used in the BCG receivers. AD events were ascertained at least 365 days after the index date to limit the inclusion of prevalent disease. Moreover, patients with a follow-up of less than one year from the index date were excluded from the study. 

Matching the HUH dataset [30] with the norms set for the other cohorts was performed by redefining the observation time according to that of the CHS data (2000–2018). Data were reanalyzed to match the same inclusion–exclusion criteria implemented for the CHS cohort, and the index date was set at the time of BC diagnosis.

In the dataset of UCLAH, the BCG-treated patients were defined as treated with at least one BCG instillation. Notably, records on additional instillations for completing a full cycle (total of six BCG instillations) are mostly missing. This exception relies on the common practice of switching from a hospital protocol to community-based clinics, which provide additional cycles of BCG admission (total along 6 weeks, a weekly BCG instillation). The index date was set as the time of the first BCG instillation for the BCG cases, and for non-BCG receivers, as the median time between BC diagnosis and the first BCG instillation in the BCG receivers. Patients with a follow-up of less than one year from the index date were excluded from the study. Patients with a history of dementia or AD (pre-BC diagnosis) were excluded on the basis of having the outcome prior to BC diagnosis.

Additional inclusion–exclusion criteria used for the PD, stroke, and T2D can be found in Appendix A.

### 2.4. Statistical Analyses

Data analysis was performed using R v3.6.1 (2019) and the *dplyr* package (v0.84, 2019) [35]. The statistical analyses were performed on the CHS dataset for the full cohort, male only, and female only, partitioned to those diagnosed with BC at the age <75 and ≥75 years old. The HUH and UCLAH datasets were examined as a full cohort only to ensure statistical robustness.

The effect of BCG treatment was estimated for every cohort and sub-cohort using Cox PH models, and the following covariates adjustment (age at diagnosis and sex) was performed using the *survminer* R package (v0.4.7, 2020) [36]. In addition, a competing risk analysis using the Fine–Gray model was conducted for measuring the primary outcome [37,38]. An alpha level of 0.05 was used in all statistical tests. For the KM survival test, we indicate the HR derived from the use of an unadjusted Cox regression analysis as HR(s). The nonparametric Wilcoxon rank test was applied for statistical hypothesis testing by comparing two samples.

The statistical analyses were performed on the CHS dataset for the PD risk examined—the full cohort, male only, and female only, partitioned to those diagnosed with BC aged <75 and ≥75. The statistical analyses were performed on the CHS dataset for the stroke risk examined—the full cohort, male only, and female only. The statistical analyses were performed on the CHS dataset for Diabetes risk examined the full cohort only.

A technical outline of the code (Appendix A) includes a description of the analyses preformed. This appendix includes a sample code used to analyze the HUH cohort’s risk of Alzheimer’s disease. This code is representative of the analyses and filtrations conducted in the CHS, HUH, and UCLAH cohorts for any type of survival disease outcome.

## 3. Results

The initial CHS dataset includes 9959 BC patients, and of these, 2600 were treated with BCG. Post-filtration, the CHS cohort comprised 6725 BC patients (1578 patients treated with BCG, and 5147 not treated with BCG), including 5558 (82.6%) males and 1167 (17.4%) females (Table 1). The filtered HUH cohort included 700 patients (408 treated with BCG and 292 untreated), with a similar bias toward males (males and females account for 83.4% and 16.6%, respectively) (Table 2). The comparison of the CHS and the HUH cohorts showed that the CHS has a smaller percentage of BCG-treated patients (23.5% vs. 58.3%). The mean follow-up time (by a 5-days resolution) in the CHS cohort was similar between the BCG and BCG-free groups (2640 (±1360) vs. 2610 (±1540)), whereas the mean follow-up time in the HUH cohort was significantly longer for the BCG group (2870 (±1840) vs. 1700 (±1400)). The UCLAH cohort from California comprised 2191 BC patients, including 132 who were treated with BCG, after applying the filtering criteria. The mean follow-up time was 3.5 years. Detailed clinical characteristics of the studied cohorts are presented in Table 1 and Table 2, and Appendix A and Appendix A.

### 3.1. Association of AD/Dementia and BCG Treatment in BC Patients

We defined the outcome as the diagnosis of the subjected disease. For AD and dementia, a total of 411 primary outcome events occurred in the CHS cohort (Table 1). Altogether, 4.8% of the patients treated with BCG developed AD, compared with 6.1% of the controls. The Kaplan–Meier (KM) estimator displayed an AD-free survival curve of the BCG recipients, which was significantly different from the survival curve of non-BCG treated patients (HR(s) of 0.734, 95% confidence interval (CI) of 0.571–0.943, and *p*-value = 0.015) (Figure 1A).

The adjusted Cox proportional hazards regression (Cox-PH) model showed a reduced AD incidence rate in the BCG-treated group, as compared with the control group (HR of 0.787, 95% CI of 0.612–1.012, and *p*-value = 0.062). A rigorous competing risk model showed a reduced AD incidence rate in the BCG-treated group relative to the control with a borderline statistical significance (HR of 0.837, 95% CI: 0.651–1.076, and *p*-value = 0.165) (Table 3).

AD and dementia are age-dependent diseases. Thus, we have stratified the CHS cohort by age partition according to the calculated median age of the BC patients (median 74 years; Appendix A). BC patients diagnosed at an age younger than 75 years-old (3625 patients) and treated with BCG showed no notable difference from the non-BCG-treated patients (Figure 1B and Table 3). Contrastingly, for patients diagnosed ≥75 years old (3100 patients), the AD-free survival curve of the BCG-treated patients was significantly different from that of non-BCG treated patients (Figure 1C). The tested statistical models showed a substantially reduced risk for AD by 30.6% (according to the adjusted Cox PH model), and 27.4% (according to the more stringent competing risk model). The results between the BCG and the control groups were significant for Cox PH and competing risk models (with *p*-values of 0.024 and 0.0473, respectively) (Table 3). Similarly, the ranges of the confidence interval (95% CI) were 0.506–0.953 and 0.529–0.996, respectively. These results confirm a reduced risk for developing AD following intravesical BCG immunotherapy.

We then tested an independent dataset of the HUH cohort (Figure 1D; Appendix A). The analysis showed profound differences in the risk of developing AD, with only 3.2% of the patients treated with BCG developing AD, compared with 5.8% of the controls. The Kaplan–Meier (KM) estimator displayed an AD-free survival curve of the BCG recipients significantly different from the survival curve of non-BCG-treated patients (HR(s) of 0.264, 95% CI: 0.126–0.559, and *p* < 0.001) (Figure 1D). The more stringent analyses that account for competing events (multivariable-adjusted Cox regression and competing risk analysis; Table 3) confirmed that BCG treatment was associated with a lower risk of developing AD (HR of 0.251, 95% CI: 0.117–0.536, and *p*-value < 0.001; HR of 0.416, 95% CI: 0.203–0.853, and *p*-value = 0.017, respectively). The proportional hazards (PH) regression for the Cox regression model was satisfied for both CSH and HUH cohorts, tested using the Schoenfeld residuals examination [39].

In order to examine the effect between the sexes, we stratified the CHS cohort by gender. As BC patients are quite male-biased, the statistical power for the analysis was somewhat limited. Still, when examining the cohort of male patients, the KM survival analysis resulted in a substantially lower HR(s) of 0.779 (95% CI: 0.592–1.024), but insignificant statistics (*p*-value = 0.074). The other tests of multivariable-adjusted Cox regression and competing risk analysis also failed to show a significant HR difference in the AD incidence rate between the BCG and control groups. However, the female cohort showed a reduced HR in three complementary tests: the Kaplan–Meier analysis (HR(s) of 0.572), the multivariable-adjusted Cox regression (HR of 0.588), and the competing risk analysis (HR of 0.639) (Figure 2, Appendix A).

In order to examine the effect of the age groups within each gender sub-group, we stratified the groups by age and gender (Appendix A) and we confirmed that most of the signal of the BCG-treated versus -untreated was associated with females ≥75 years. The results were statistically significant for several complementary tests.

In order to test the relevance of our findings in a different medical “milieu,” we analyzed the UCLAH cohort (4760 BC patients with 132 AD cases; Appendix A). Notably, there were no reports of AD cases among the 315 BCG-treated patients, even though the median age of diagnosis was 69 years old (Appendix A). We performed hypergeometric hypothesis testing to test whether we can reject the null hypothesis that no AD cases among BCG-treated sets are due to random sampling. The calculated *p*-value was 0.008, providing further support for the main observation of a favorable outcome for the BCG treatment in BC patients.

### 3.2. Association of TB BCG Vaccination Regimens and AD Prevalence

We further examined the correlation of people that received intradermal BCG, as a vaccine against TB and the risk of AD. The test aimed to reveal whether the BCG that was given as a vaccine against TB at a population level has a long-term effect from early life vaccination to elderly AD. To this end, we examined the correlation between varying BCG regimens and AD prevalence in European countries. The BCG coverage data extraction and AD prevalence by age was collected as described before [40]. Analyzing the correlation between the BCG vaccination population coverage (in many European countries) and the percentage of AD patients within the total population resulted in a negative linear correlation (Appendix A). By testing the entire population, females and males, only the male subpopulation resulted in significant results (full cohort: R = −0.36, *p*-value = 0.08; females: R = −0.35, *p*-value = 0.098; males: R = −0.43, *p*-value = 0.035). However, a refined analysis performed on the male cohort found no significant correlation between the BCG population coverage weighted by the percentage of AD patients within age groups (30–59 and ≥60). For the males age group, 30–59 years: R = 0.1, *p*-value = 0.63; for the ≥60 age group: R = −0.05, *p*-value = 0.81 (Appendix A).

### 3.3. Association of PD and BCG Treatment in BC Patients

Results from the three different cohorts and several statistical tests confirmed the relevance of BCG treatment for AD and dementia neuroprotection. Stimulated from the findings that showed BCG to be neuroprotective in an experimental mouse model of PD [41], we extended our hypothesis to test the impact of BCG treatment on PD, another major age-dependent neurodegenerative disease. In the CHS cohort, a total of 6766 patients, 1669 were treated with BCG and 5097 were not treated with BCG (Appendix A). During follow-up, PD was diagnosed in 33 patients treated with BCG (1.98%), and in 153 patients not treated with BCG (3.0%). The results proved significant, in both the Cox PH and competing risk analyses. However, as the number of patients is quite limited, the 95% CI ranges broadly (HR = 0.682, 95% CI: 0.468–0.994, and *p*-value = 0.047; competing risk results: HR = 0.716, 95% CI: 0.491–1.044, and *p*-value = 0.082) (Figure 3 and Table 4). Thus, despite the relatively limited number of cases, a beneficial PD outcome to BCG-treated BC patients was a reduction in the risk of developing PD by at least 28%.

### 3.4. Association of Stroke and T2D and BCG Treatment in BC Patients

As BCG has been shown to significantly correlate with a lower risk of neurodegenerative disorders with the underlying mechanism most likely being immunological, we examined whether nonimmunological, age-dependent diseases could control for our observational results (Figure 1, Figure 2 and Figure 3). To this end, we examined whether BCG treatment in BC patients is associated with a reduced risk of stroke in the CHS cohort. Both the Kaplan–Meier analysis performed and the multivariate cox regression were found insignificant, and the HR remained mostly constant (HR = 1.048, 95% CI: 0.846–1.229, and *p*-value = 0.667; HR = 1.068, 95% CI: 0.861–1.326, and *p*-value = 0.547, Appendix A). Results conducted with sex stratification remained insignificant (Appendix A).

While causality is impossible to infer from such a retrospective analysis, we examined whether other chronic and major diseases would be affected similarly by BCG treatment. This led us to examine whether patients treated by BCG would also have an altered risk of Type 2 diabetes (T2D), a common metabolic disease that is strongly linked to the population BMI. Examining the risk of developing diabetes at follow-up time failed to yield a significant association between the disease and BCG treatment. The KM analysis performed on the population defined in a Cox PH analysis also showed no significant results (HR = 0.710, 95% CI: 0.429–1.490, and *p*-value = 0.482). Similar results were also demonstrated in a log-rank test (log rank: chi-square 0.5, df = 1, *p* value = 0.5) and the Cox regression with covariate adjustment (HR = 0.890, 95% CI: 0.4785–1.679, and *p*-value = 0.732). The results are summarized in Appendix A.

## 4. Discussion

The BCG vaccine, originally used against tuberculosis (TB), has been used for several decades as a first-line therapy for preventing BC recurrence. Beyond those uses, BCG was shown to nonspecifically modulate the immune system [40,42,43]. In this retrospective study, following three independent cohorts, the majority of the observational evidence suggests that BCG treatment in BC patients is associated with a significantly reduced risk of AD.

The percentage of patients treated with intravesical instillation of BCG following a TURBT treatment varied between cohorts. For the CHS, the BCG-treated patients constituted 23% of the cohort, whereas the HUH and UCLAH BCG-treated patients constituted 58% and 6% of their respective cohorts. As BCG therapy is the most common first-line therapy for NMIBC patients [44], it is not surprising for a tertiary hospital to propose it as a preferred treatment. In contrast, the small percentage of patients treated with BCG in the UCLAH cohort (6%) reflects the underuse of this treatment in the United States [45]. AD was diagnosed in 6.1% of the CHS patients in the study, similar to the proportion reported in the general Israeli population [46]. However, in the HUH and UCLAH cohorts, the proportion diagnosed with AD was significantly lower (4.3% and 3.4%, respectively), though not much lower than the AD estimate from a large European meta-analysis (4.7%) [47]. These lower rates could reflect an inverse correlation between cancer and AD [48,49,50]. While cancer patients are still under treatment, it is reasonable to expect that caregivers may delay diagnosis tests for other major diseases (e.g., AD/dementia), leading to surveillance bias [51]. Lastly, as AD is characterized by years of gradual cognitive decline, the relatively shorter mean follow-up time in the UCLAH cohort may lead to survival bias, favoring better fit patients than frailer ones [48].

Though within the whole CHS cohort, BCG-treated patients showed a reduced risk of AD, the inherent differences in AD risk by sex and age [14,52] led us to examine the data accordingly. Stratifying the CHS population by the median age of BC patients (≥75 at diagnosis, <75 at diagnosis) showed a significant correlation in both the Cox PH model and competing risk analyses between a reduced AD risk and BCG administration (Figure 1 and Table 3). This may indicate that the population of ≥75 could be uniquely affected by the BCG treatment. The early onset form of AD is thought to have a strong genetic component and a different etiology [53]. As cases of early AD onset are included in the <75 diagnosis-year group, the cohort may be etiologically different, providing a possible explanation for the observed differences by the age partition.

Sex stratification of the CHS population demonstrated a statistically insignificant AD risk after BCG administration (Appendix A). The female cohort showed a further reduced HR in comparison to the male cohort, and this may be a result of the expected higher rate of female AD cases within the group (Appendix A) [54]. Those differences may also be a result of inherent sex differences in AD, such as a longer life expectancy for women, hormonal differences, and differential cognitive performance [55,56]. Further studies on larger cohorts are needed to validate a sex-related effect.

The lower risk of developing AD in BCG-treated patients suggests that the bladder instillation of BCG may cause a systemic immune response that reduces the risk of AD [51]. This is consistent with results showing that, several weeks post-intravesical BCG instillation, the levels of the IL-2 cytokines in the serum are increased tenfold [57], which may lead to an increase in the beneficial immunosuppressive Treg population [29,57]. Therefore, the BCG-modulated systemic immune response may prevent or slow the development of clinical definite syndrome as in the case of multiple sclerosis [58]. Additionally, the BCG vaccination of adults induces a systemic shift in glucose metabolism from oxidative phosphorylation to aerobic glycolysis, a state of high glucose utilization [59] that has been shown to restore youthful immune functions and reverse cognitive ageing [60].

BCG, when used as a vaccine to prevent TB, was not found to be associated with a reduced risk of AD (Appendix A). BCG immunization against TB is mostly administered at birth (exceptions are medical personnel that are vaccinated as adults). The beneficial nonspecific effect of BCG vaccination is thought to have a lasting effect of approximately 20 years. As AD is prevalent at an old age, it is not surprising that countries with higher BCG vaccination rates did not support an association with the AD reduced rate. In the US population where BCG vaccination is considered for only highly selected people (children and health workers in contact of active tuberculosis cases), very few of the bladder cancer patients had probably ever received BCG. The vaccination statuses of the Israeli cohorts are markedly different. BC patients of the Israeli cohorts were vaccinated at birth as BCG vaccination was mandatory in Israel up to 1982 [61] and as a large part of new immigrants, especially from the former Soviet Union, were also subjected to a compulsory BCG vaccination at birth. BCG efficacy to prevent TB or leprosy is considered to weaken after 15–20 years [2,3] and, therefore, the status of previous BCG vaccinations should not have much influence in a population of BC patients (median age is 74). Nevertheless, two reports showed protection against TB for more than 40 years [62,63]. Our data do not support a boosting effect by the BCG immunotherapy treatment. The age partition of the tested cohorts shows that the significant effect on AD/dementia was mostly attributed to the subpopulation at age > 75, where the effect on patients <70 years was negligible. We argue that the early sensitization to BCG does not appear to have a role on its effect in elderly BC-treated patients who benefited from a reduced risk of developing AD and PD.

Several studies have indicated a connection between immune system dysfunction and neurodegenerative diseases such as multiple sclerosis [58,64] and PD [65,66] with a protective role of BCG-induced Tregs [41]. Favorably modulating the immune system post-BCG administration may potentially affect neuronal inflammatory responses that underlie PD, possibly delaying symptoms onset. Indeed, we showed the same trend with a reduced risk for PD ranging between 31.8% and 28.4% (Figure 3 and Table 4). The statistical power of the PD was limited. The limited number of PD patients that complied with the inclusion–exclusion criteria reflects the earlier age of diagnosis of PD patients, which often occurs prior to the BC diagnosis. While AD diagnosis is associated with a gradual decline in functionality that can be traced back many tears, PD diagnosis is often visible by motor dysfunction; thus, the time between symptoms to diagnosis is much shorter relative to AD [67].

The study has several limitations precluding a definite conclusion (causality) regarding the effect of BCG on AD risk. First, an inherent bias exists due to the design of the study that examines AD risk in BC patients. Several studies have linked a reduced risk for AD in cancer patients [49,50,51]. Nonetheless, in this study, all cohorts examined were diagnosed with BC whether intravesical BCG treated or not. It is thus safe to hypothesize that those ramifications would similarly affect the treatment and control groups. Second, confounding-by-indication for BCG treatment may exist within the NMIBC patients. BCG treatment is administered by urologists to patients they see fit to undergo this unpleasant, repeated treatment with its potential side effects. It cannot be ruled out that frailer patients were less likely to be administered BCG, resulting in a difference between the BCG and control populations. Thirdly, inherent differences for the protective effects among cohorts and even within a cohort over time are attributed to the variability between the use of different BCG strains [68], information that was not available in the studied BC cohorts. Finally, there is an inherent bias due to the high number of censored patients and lack of long-term follow-up in some cases.

In summary, we showed that in three separate cohorts, two continents, and three different clinical settings, intravesical BCG treatment is associated with a reduced AD risk in BC patients. The results complied with the statistical norm of competing risk analyses that mitigated the effect of deaths within the different examined groups. Current prospective studies in elderly people inoculated with intradermic BCG [69,70], and several ongoing clinical trials (e.g., Clinical Trials Identifier: NCT04507126) should be extended to evaluate the potential protective effect of BCG against AD and PD. The prevalence of major neurodegenerative diseases is constantly increasing. Specifically, between 2000 and 2018, deaths resulting from stroke and heart disease decreased, whereas reported deaths from AD increased 146% [15]. This study exposes an association for a favorable outcome by BCG treatment. Importantly, BCG is an approved vaccine that was used successfully for almost 100 years to control TB across the globe. The numerous nonspecific effects were acknowledged in view of other viral infections [40,71] and a range of pathologies [1,72,73]. The potential of the stimulation of the immune system in the elderly with a live vaccine to provide protection against neurodegenerative diseases is an attractive translational research route. Delaying the onset of these major neurodegenerative diseases (AD/dementia and PD) or suppressing their progression await validation from ongoing clinical trials. Still, even a small beneficial affect will have a significant impact on the health and quality of life of the elderly in our aging society.

## 5. Conclusions

In this study, we highlighted the nonspecific immune effects of the BCG vaccine and BCG immunotherapy, showing that it may trigger beneficial immunomodulation in neurodegenerative disorders. In three separate cohorts, we showed that intravesically BCG treatment is associated with a reduced AD risk in BC patients with the association attributed mainly to the older sub-cohort (≥75). Our results, and the accumulating evidence indicating an association between BCG and a reduced risk of AD and PD call for random clinical trials to examine those associations. Any potential treatment, even partially reducing the rate of those diseases, would extensively benefit neurodegenerative disease patients and society as a whole.

## Figures and Tables

**Figure 1 vaccines-09-00491-f001:**
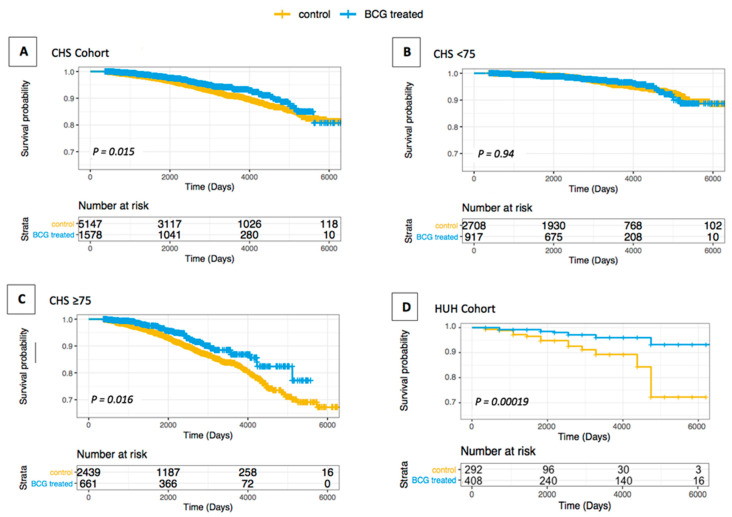
Kaplan–Meier survival curves of AD-free bladder cancer patients treated or not treated with BCG. The whiskers on the Kaplan–Meier (KM) survival plots represent the censored patients. (**A**) CHS cohort; (**B**) CHS < 75 years old; (**C**) CHS ≥ 75 years old; (**D**) HUH Cohort.

**Figure 2 vaccines-09-00491-f002:**
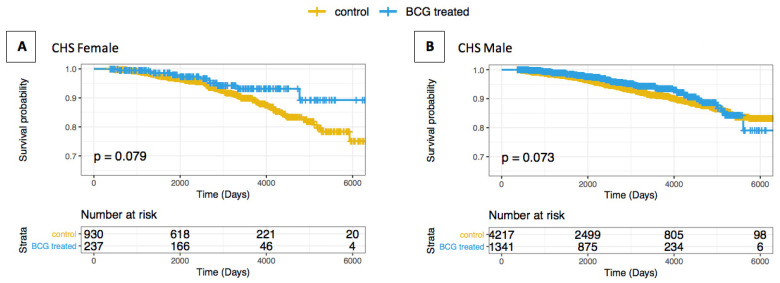
Kaplan–Meier survival curves of bladder cancer patients treated or not treated with BCG by gender. The whiskers on the Kaplan–Meier (KM) survival plots represent the censored patients. (**A**) CHS female AD-free cohort; (**B**) CHS male AD-free cohort.

**Figure 3 vaccines-09-00491-f003:**
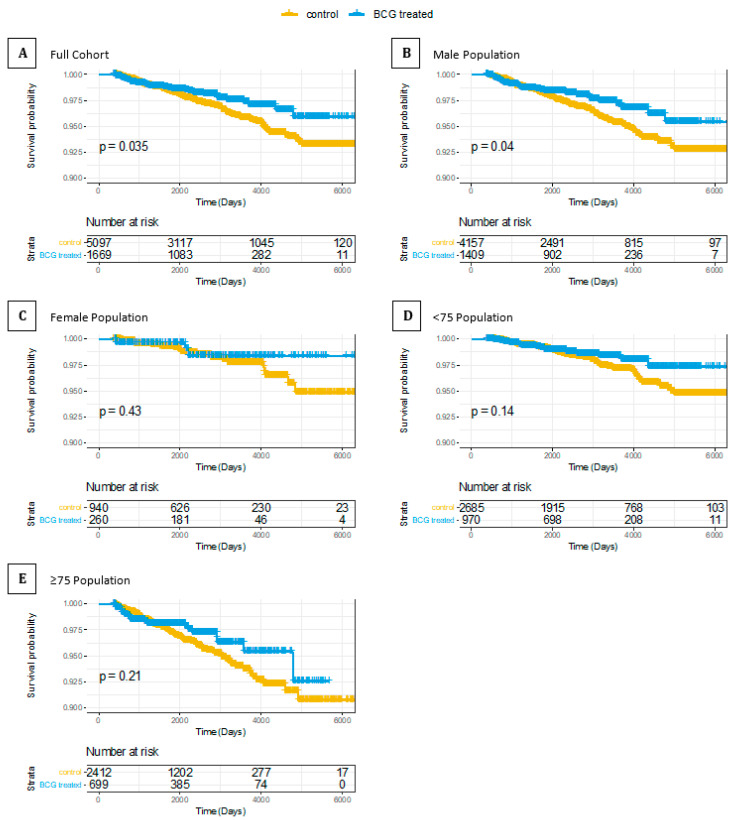
KM survival curve of bladder cancer patients treated or not treated with BCG, examined for PD risk. The whiskers on the KM survival plots represent the censored patients. (**A**) Full CHS cohort; (**B**) male only; (**C**) female only; (**D**) <75 population. (**E**) ≥75 population.

**Table 1 vaccines-09-00491-t001:** Characteristics of patients in the CHS population.

Strarificaiton Criteria	Group	Non-BCG (N = 5147)	BCG (N = 1587)	Overall (N = 6725)
Gender	Male	4217 (81.9%)	1341 (85.0%)	5558 (82.6%)
Gender	Female	930 (18.1%)	237 (15.0%)	1167 (17.4%)
AD Diagnosis	Non-AD	4811 (93.5%)	1503 (95.2%)	6314 (93.9%)
AD Diagnosis	AD	336 (6.5%)	75 (4.8%)	411 (6.1%)
Age at TCC Diagnosis (years)	Mean (SD)	73.9 (8.11)	72.8 (7.54)	73.7 (7.99)
Age at TCC Diagnosis (years)	Median [Min, Max]	74.0 [60.0, 104]	73.0 [60.0, 99.0]	74.0 [60.0, 104]
Stage	<T2	4655 (90.4%)	1415 (89.7%)	6070 (90.3%)
Stage	≥T2	492 (9.6%)	163 (10.3%)	655 (9.7%)
Age of Death (years)	Mean (SD)	82.8 (8.11)	82.2 (7.65)	82.2 (8.03)
Age of Death (years)	Median [Min, Max]	83.0 [61.0, 110]	83.0 [62.0, 103]	83.0 [61.0, 110]
Age of Death (years)	Missing	2036 (39.6%)	824 (52.2%)	2860 (42.5%)
Follow-Up Time (days)	Mean (SD)	2610 (1540)	2640 (1360)	2610 (1500)
Follow-Up Time (days)	Median [Min, Max]	2390 [366, 6570]	2520 [368, 6490]	2420 [366, 6570]

Non-BCG are BC patients not treated with BCG post-filtration (control group). BCG are BC patients treated by BCG instillation. The statistics of the cohorts is post-filtration according to the inclusion–exclusion criteria. AD, Alzheimer’s disease; TCC, transitional cell cancer; Stage, bladder cancer stage.

**Table 2 vaccines-09-00491-t002:** Characteristics of patients in the HUH population.

Strarificaiton Criteria	Group	Non-BCG (N = 292)	BCG (N = 408)	Overall (N = 700)
Gender	Male	234 (80.1%)	350 (85.8%)	584 (83.4%)
Gender	Female	58 (19.9%)	58 (14.2%)	116 (16.6%)
AD Diagnosis	Censor	151 (51.7%)	222 (54.4%)	373 (53.3%)
AD Diagnosis	AD	17 (5.8%)	13 (3.2%)	30 (4.3%)
AD Diagnosis	Death	124 (42.5%)	173 (42.4%)	297 (42.4%)
Age at TCC Diagnosis (years)	Mean (SD)	74.6 (8.11)	73.9 (8.09)	74.2 (8.10)
Age at TCC Diagnosis (years)	Median [Min, Max]	74.0 [60.0, 98.0]	73.0 [60.0, 98.0]	73.0 [60.0, 98.0]
Age of Death (years)	Mean (SD)	79.9 (7.72)	82.4 (7.93)	81.4 (7.93)
Age of Death (years)	Median [Min, Max]	79.5 [61.0, 102]	83.0 [64.0, 101]	82.0 [61.0, 102]
Age of Death (years)	Missing	168 (57.5%)	235 (57.6%)	403 (57.6%)
Follow-Up Time (days)	Mean (SD)	1700 (1400)	2870 (1840)	2380 (1770)
Follow-Up Time (days)	Median [Min, Max]	1100 [365, 6570]	2560 [365, 6570]	1830 [365, 6570]

Non-BCG are BC patients not treated with BCG post-filtration (control group). BCG are BC patients treated by BCG instillation. The statistics of the cohorts is post-filtration according to the inclusion–exclusion criteria. AD, Alzheimer’s disease; TCC, transitional cell cancer.

**Table 3 vaccines-09-00491-t003:** Risk assessment of AD in CHS and HUH cohorts by Kaplan–Meier (KM), Cox proportional hazards regression (Cox), and competing risk analysis (CR).

Group	Analysis	HR (95% CI)	*p*-Value
CHS Full Cohort	KM	0.734 (0.571–0.943)	0.015
CHS Full Cohort	Cox	0.787 (0.612–1.012)	0.062
CHS Full Cohort	CR	0.837 (0.651–1.076)	0.165
CHS < 75	KM	0.984 (0.650–1.490)	0.939
CHS < 75	Cox	0.996 (0.637–1.464)	0.87
CHS < 75	CR	1.036 (0.684–1.568)	0.869
CHS ≥ 75	KM	0.679 (0.495–0.931)	0.016
CHS ≥ 75	Cox	0.694 (0.506–0.953)	0.024
CHS ≥ 75	CR	0.726 (0.529–0.996)	0.047
HUH Full Cohort	KM	0.264 (0.126–0.559)	0.001 *
HUH Full Cohort	Cox	0.251 (0.117–0.536)	0.001 *
HUH Full Cohort	CR	0.416 (0.203–0.853)	0.017

Full cohorts of CHS and HUH include patient age ≥ 60; CHS < 75 include patients of 60 ≤ age < 75; CHS < 75 include patient aged ≥ 75. HR, hazard ratio; KM, The HR derived from the use of an unadjusted Cox regression analysis for Kaplan–Meier (KM) survival analysis; Cox, Cox proportional hazards regression; CR, Fine–Gray competing risk model. *p*-values marked with * are <0.001.

**Table 4 vaccines-09-00491-t004:** Risk assessment of PD in CHS by Kaplan–Meier (KM), Cox proportional hazards regression (Cox), and competing risk analysis (CR).

Group	Analysis	HR (95% CI)	*p*-Value
CHS full PD	KM	0.668 (0.459–0.974)	0.036
CHS full PD	Cox	0.682 (0.468–0.994)	0.047
CHS full PD	CR	0.714 (0.491–1.044)	0.082
CHS PD M	KM	0.656 (0.458–0.941)	0.022
CHS PD M	Cox	0.708 (0.493–1.016)	0.06
CHS PD F	KM	0.878 (0.329–2.339)	0.794
CHS PD F	Cox	0.927 (0.346–2.485)	0.88
CHS PD < 75	KM	0.629 (0.369–1.070)	0.087
CHS PD < 75	Cox	0.646 (0.379–1.100)	0.107
CHS PD ≥ 75	KM	0.790 (0.510–1.225)	0.292
CHS PD ≥ 75	Cox	0.819 (0.528–1.272)	0.375

Full cohorts of CHS include patients aged ≥60 diagnosed with PD; CHS PD M includes male patients aged ≥60 diagnosed with PD; CHS PD F includes female patients aged ≥60 diagnosed with PD. CHS PD < 75 includes patients of 60 ≤ age < 75 diagnosed with PD; CHS PD ≥ 75 includes patients aged ≥75 diagnosed with PD. HR, hazard ratio; KM, the Cox regression analysis for Kaplan–Meier (KM) survival analysis; HR derived from the use of an unadjusted cox regression analysis (HR(s)); Cox, Cox proportional hazards regression; PD, Parkinson’s disease.

## Data Availability

No applicable.

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
