# Peer review of "Bladder Cancer Immunotherapy by BCG Is Associated with a Significantly Reduced Risk of Alzheimer’s Disease and Parkinson’s Disease"

_vaccines, 2021, doi:10.3390/vaccines9050491_

Round 1
Reviewer 1 Report
Bladder Cancer Immunotherapy by BCG…
Overview
The paper reports a retrospective study to determine whether repeated doses of the BCG vaccine during treatment for bladder cancer can provide relief from Alzheimer’s disease (AD) and from Parkinson’s disease (PD). Data from three different medical centers are studied with regard to onset of AD or PD; analysis being done using a proportional hazards model.
The medical centers involved were Clalit Health Service (CHS), Hadassah University Hospitals (HUS) and UCLA Health systems (UCLAH). The bulk of the results in the main paper refer to CHS.
Detailed comments
- The criteria for including patients in the study conform to common-sense guidelines, but this reviewer is not sure whether these guidelines completed or not.
- The data analysis was performed using standard packages in R and there are no obvious shortcomings in the reported results. However, it might be appropriate to include a short technical appendix describing the methodology? I am not sure how familiar readers of the journal would be with the methods employed?
- The statistical power of some of the analyses has to be questioned. Given that there are 3 centers, and that the data are coded by age and sex it would be interesting to conduct an analysis of the complete data set, with main effects, and interactions as needed. I recognize that the UCLAH data could only be included if it is assumed that there is no “center effect”, but it is worth exploring.
Author Response
- The criteria for including patients in the study conform to common-sense guidelines, but this reviewer is not sure whether these guidelines completed or not.
Reply: With respect to ethical consideration (e.g. patient privacy and consents), the research was conducted according to the guidelines and approval of all institutions (as reported in Methods). For the selection of inclusion-exclusion criteria, we benefit from the guidance of urologist, epidemiologists and experts of electronic health records data extraction.
- The data analysis was performed using standard packages in R and there are no obvious shortcomings in the reported results. However, it might be appropriate to include a short technical appendix describing the methodology? I am not sure how familiar readers of the journal would be with the methods employed?
Reply: Thank you for your comment. Based on your recommendations, we added to the revised version a thorough technical appendix describing the methodology used. This detailed technical log will help users to view the protocol used for presenting the results.
- The statistical power of some of the analyses has to be questioned. Given that there are 3 centers, and that the data are coded by age and sex it would be interesting to conduct an analysis of the complete data set, with main effects, and interactions as needed. I recognize that the UCLAH data could only be included if it is assumed that there is no “center effect”, but it is worth exploring.
Reply: Thank you for your comment. Unfortunately, due to separate institutional regulations that prohibit data sharing, we were unable to share or combine the data for such an analysis. Instead we were able to show that despite some differences in the cohorts, they can be considered as ‘independence’ cohorts. Obtaining a similar trend towards AD/ dementia (albeit, at varying statistical significance) serves as a support to the main finding on BCG treatment.
Reviewer 2 Report
Estimated Authors,
I've read with interest the present paper presented by the study group lead by Klinger and reporting on the potential preventive effect of BCG vaccination on Alzheimer's disease.
Because of the relatively low cost of BCG vaccine, and the very high impact of AD on the worldwide healthcare systems (not mentioning the economic burden) it is quite obvious that even small preventive effects may be of seminal importance.
However, it should be keep in mind (as the Authors have done in their discussion section) that:
1) BCG vaccine is among the most commonly delivered vaccinations worldwide;
2) AD is among the most commonly reported diseases of the elderly, particularly in High income countries;
3) to date, actual aetiology of AD remains relatively unclear, and potential risk factors have not been completely elucidated. For instance, the greatest risk factors for late‐onset Alzheimer's are older age, genetics, and having a family history of Alzheimer's (that in turn stresses the role of underlying genetics or some behavioural and still not identified factors).
Such introduction to stress that - albeit of certain interest, such intrinsic limits must be clearly addressed by the study Authors, more extensively than in the present version of the paper. this is particularly significant as the study population is supposedly quite heterogeneous in terms of ethnic composition (i.e. 1 american center vs. 2 Israeli centers, all of them encompassing a population that cannot be easily defined as homogenous in terms of genetic background).
in summary, I warmly recommend the authors to:
1) address the aforementioned intrinsic limits with a more extensive description;
2) include a more extensive description of the main characteristics of the study (e.g. now Table 1 includes only CHS cohort, please include the whole of sampled patients)
3) please be aware that the vaccination policies against BCG are quite heterogenous around the world; as a consequence (particularly when dealing with population assisted by Israeli centers) the vaccination schedule received by study participants may be in turn inconsistent among the very same study population - also this topic must be properly addressed. If it possible, Authors should report in the main text available data on the latency between vaccination (also in broader terms, as childhood vs. adulthood) in the sampled participants.
Some further recommendations:
1) Table 1 is affected by some formatting issue (particularly row of Stage: please fix it)
2) Figure 1 to 3 may be ambiguous. For example: Figure 3e seemly show a strikingly difference in the assessed outcome between the two study groups, but not coincidentally the correspondent p value was estimated to 0.21. In facts, the scale of Y axis ranges between 0.9 and 1.0, with a subsequent amplification of the differences. Please change the range to 0.0 to 1.0
Author Response
I've read with interest the present paper presented by the study group lead by Klinger and reporting on the potential preventive effect of BCG vaccination on Alzheimer's disease.
Because of the relatively low cost of BCG vaccine, and the very high impact of AD on the worldwide healthcare systems (not mentioning the economic burden) it is quite obvious that even small preventive effects may be of seminal importance.
However, it should be keep in mind (as the Authors have done in their discussion section) that:
1) BCG vaccine is among the most commonly delivered vaccinations worldwide;
2) AD is among the most commonly reported diseases of the elderly, particularly in High income countries;
3) to date, actual aetiology of AD remains relatively unclear, and potential risk factors have not been completely elucidated. For instance, the greatest risk factors for late‐onset Alzheimer's are older age, genetics, and having a family history of Alzheimer's (that in turn stresses the role of underlying genetics or some behavioural and still not identified factors).
Reply: Thank you for your comment. Based on your recommendations, we added to the introduction more ‘background’ knowledge on AD and on the BCG vaccination. In the revised version these paragraphs are marked in red (see revised introduction).
Such introduction to stress that - albeit of certain interest, such intrinsic limits must be clearly addressed by the study Authors, more extensively than in the present version of the paper. this is particularly significant as the study population is supposedly quite heterogeneous in terms of ethnic composition (i.e. 1 american center vs. 2 Israeli centers, all of them encompassing a population that cannot be easily defined as homogenous in terms of genetic background).
in summary, I warmly recommend the authors to:
- address the aforementioned intrinsic limits with a more extensive description;
Reply: Thank you for your suggestion. Based on your recommendations, we added to the revised version introduction a more extensive description of the aforementioned intrinsic limits. While the familial genetic basis for AD (leading to early-onset) are knowns as they occur at an early age, they are not affecting our analysis. Nevertheless, we explain the important issue of known allele as risk factors and explain the complexity of the etiology of the disease (Revised Page 4-5)
We also included the missing references to support some of the genetic data that are in the background of our analysis.
2) include a more extensive description of the main characteristics of the study (e.g. now Table 1 includes only CHS cohort, please include the whole of sampled patients)
Reply: As suggested we moved the Table of the cohorts from the supplemental material to the main text (Revised Tables are 1-4). The new Table for HUH cohort characteristics is currently Table 2. A furthermore in-depth stratification of all three cohorts can be found in the supplemental material Tables S1-S3 and S5-S8.
- please be aware that the vaccination policies against BCG are quite heterogenous around the world; as a consequence (particularly when dealing with population assisted by Israeli centers) the vaccination schedule received by study participants may be in turn inconsistent among the very same study population - also this topic must be properly addressed. If it is possible, the Authors should report in the main text available data on the latency between vaccination (also in broader terms, as childhood vs. adulthood) in the sampled participants.
Reply: Based on your recommendations, we added to the revised version a more extensive discussion of the matter with respect to vaccination policy and the unlikely of the treatment of BC patients to ‘boost’ the birth/childhood TB vaccination (page 14).
Some further recommendations:
- Table 1 is affected by some formatting issue (particularly row of Stage: please fix it)
- Reply: We solved the formatting issues in the edited version for Tables 1-2.
- Figure 1 to 3 may be ambiguous. For example: Figure 3e seemly show a strikingly difference in the assessed outcome between the two study groups, but not coincidentally the correspondent p value was estimated to 0.21. In facts, the scale of Y axis ranges between 0.9 and 1.0, with a subsequent amplification of the differences. Please change the range to 0.0 to 1.0
Reply: Thank you for your comment. For comparison matters, we kept the same scale for similar cohorts. The y axis scale in the survival analyses is produced by the data- a larger scale (from 0.0-1.0) would not have enabled the reader to see subtle differences between the different curves- hence, the scale is based on the decline observed in the data.
More specifically, in Figure 3e, what seems to be a very significant difference between the control and BCG groups, is actually driven by a low number of participants (as can be seen in the “at-risk” table in the figure. Note that many patients were censored early on. The statistical considerations are used exactly for taking into account any differences in the profiles of the BCG and control groups throughout the analysis, avoiding ‘visual’ misleading interpretations. Specifically, for Figure 3e it ended up to be overall an insignificant result. This is exactly the reason why both analyses are presented: the survival curve is seen and the statistical significance is reported, in order for the reader to be able to have an integrated and full view of the results, that includes subtle differences as well as the overall statistically solid result.
Reviewer 3 Report
The work is well done and includes aspects that could impact on everyday’s practise. The methodological approach is appropriate and the discussion is exhaustive.
“we hypothesized….PS” shift this sentence from results to method section.
Author Response
Comments and Suggestions for Authors
The work is well done and includes aspects that could impact on everyday’s practise. The methodological approach is appropriate and the discussion is exhaustive.
“we hypothesized….PS” shift this sentence from results to method section.
Reply: Thank you for your comment. Based on your recommendations, we removed from the revised version the mentioned paragraph in order to avoid redundancy (it is present as a motivation paragraph in the end of the introduction).
Round 2
Reviewer 1 Report
Your responses to items 1 and 3 are accepted. However, the supplementary materials as submitted did not include a technical appendix.
Author Response
Thank for the comment. We added the technical report as supplemental material (as asked by the editor). We refer to it in the text as a Supplemental technical appendix (PDF)

Reviewer 2 Report
Authors have addressed all my concerns, either through text's amendments or the rebuttal letter.
Therefore, I endorse the acceptance of this paper.
Author Response
We want to thank the referee for the careful reading and numerous useful suggestions.